# From RNA World to SARS-CoV-2: The Edited Story of RNA Viral Evolution

**DOI:** 10.3390/cells10061557

**Published:** 2021-06-20

**Authors:** Zachary W. Kockler, Dmitry A. Gordenin

**Affiliations:** Genome Integrity and Structural Biology Laboratory, National Institute of Environmental Health Sciences, US National Institutes of Health, Durham, NC 27709, USA; zachary.kockler@NIH.gov

**Keywords:** RNA world theory, messenger RNA, viral RNA, genome stability, viral evolution, hypermutation, APOBEC, ADAR, RNA editing

## Abstract

The current SARS-CoV-2 pandemic underscores the importance of understanding the evolution of RNA genomes. While RNA is subject to the formation of similar lesions as DNA, the evolutionary and physiological impacts RNA lesions have on viral genomes are yet to be characterized. Lesions that may drive the evolution of RNA genomes can induce breaks that are repaired by recombination or can cause base substitution mutagenesis, also known as base editing. Over the past decade or so, base editing mutagenesis of DNA genomes has been subject to many studies, revealing that exposure of ssDNA is subject to hypermutation that is involved in the etiology of cancer. However, base editing of RNA genomes has not been studied to the same extent. Recently hypermutation of single-stranded RNA viral genomes have also been documented though its role in evolution and population dynamics. Here, we will summarize the current knowledge of key mechanisms and causes of RNA genome instability covering areas from the RNA world theory to the SARS-CoV-2 pandemic of today. We will also highlight the key questions that remain as it pertains to RNA genome instability, mutations accumulation, and experimental strategies for addressing these questions.

## 1. Introduction

One of the favorite aphorisms in modern biology is the title of the Theodosius Dobzhansky’s essay “*Nothing in Biology Makes Sense Except in the Light of Evolution*” [1]. In his Synthetic Theory of Evolution, Dobzhansky defined two major factors—genetic variation (i.e., mutation and other types of genome instability) and natural selection [2]—that interplay in generating new species. Remarkably high instability levels of RNA genomes accelerate speciation to the levels that often allow documenting evolution in real time. Besides the unique opportunity for researchers, this, at times, represents a considerable threat to the species hosting RNA viral genomes, including ongoing pandemic of SARS-CoV-2. The latter resulted in recent boom of attention to mechanisms of genome instability in RNA viruses.

In today’s biological systems, the genetic material making up the genome is primarily DNA. In contrast, a plethora of viruses that infect cellular hosts throughout all kingdoms rely upon RNA as their primary genetic material. Whichever the genetic material of the virus genome, there is the requirement that the genome remains stable to allow for the transmission of viable genetic material to progeny, and to prevent the extinction of species [3,4,5,6]. However, non-catastrophic levels of genome instabilities are instrumental for accumulating beneficial variants to prepare a species to meet the challenges of ever-changing environments and allow for downstream evolution [7,8,9,10]. Therefore, a balance between a stable genome and instances of genome instability must be met.

To date, there has been numerous studies into the stability/instabilities of DNA genomes, but the same level of research has not been performed for RNA. This disparity is important to note because RNA genomes are predicted to be a vital key to evolution, as prior to the last universal common ancestor (LUCA) the RNA world theory predicts the existence of protocells that used RNA as a genetic material. Further, RNA was also proposed to be used as an enzyme to mediate all metabolic functions (as proteins had not yet evolved) [11,12,13,14,15,16,17,18]. With the reliance upon RNA for the genetic material, as well as for cellular function, there must have been efficient replication of RNA within the cell, but the modes of replication of these protocells are not known. To get a better picture for how these protocells replicated, along with their likely sources of genome instability, remnants of the mechanisms of protocell replication may remain within the genomes of modern RNA viruses.

The modes of replication of modern viruses are known (discussed below in Section 2) and have been found to have the highest mutation rates per nucleotide among all biological species [19]. Viral RNA genomes are not as stable as DNA genomes, and this could be due to multiple factors including; special features of RNA genomes, RNA virus replication machinery, high selection pressure, and the susceptibility of viral RNA to environmental and/or endogenous lesions [20,21]. Thus, these instabilities of RNA virus genomes in turn should speed up their evolution. These evolutionary insights are especially important in the light of the current (at time of publishing) COVID-19 pandemic caused by the RNA Coronavirus SARS- CoV-2. Just how SARS-CoV-2 evolved to become transmissible between humans is not yet known but would require the introduction of variants into the genome through non-catastrophic events of genome instability to gain an evolutionary advantage [22,23]. How these key variants were introduced remains unclear, but in this review, we discuss possible and likely sources of genome instability that introduced these variants as well as highlight key remaining questions from the RNA world to SARS-CoV-2 pandemic.

## 2. Replication of RNA Genomes

### 2.1. RNA Protocell Replication

The first protocells of the RNA world theory had to replicate their genome to pass the genetic material on to progeny, but the mechanism of how they replicated remains unclear. At that time, there was an inability to transcribe proteins, so the current mechanisms that organisms use to replicate their genome would not be available. Alternatively, replication of protocell’s genome potentially could proceed via a ribozyme, but this would require the presence of two RNA molecules per cell, which is unlikely in the very first protocells, as having two copies of RNA -one for the genome and one for the actual replicase ribozyme- arising independently in the same early protocell is low [13,24], so, prior to ribozyme evolution, there must have been a mechanism for replicating RNA independent of ribozymes. However, the development of a model for non-enzymatic replication of RNA genomes is at best in its infancy. Specifically, the problem is that, to date, a long-tract non-enzymatic RNA replication mechanism in nature has yet to be found (reviewed in [11]), which is further compounded by the difficulty to separate long stretches of replicated RNA strands, should long tract RNA be synthesized. This inability lends to the likelihood of a hypothesis that, instead of a long tract synthesis, a shorter type of synthesis is utilized [13,25]. This idea has its own problems based on the ends of the replicated genome. A non-enzymatic replication would be required to begin at one end and continue through the other end, which would require an improbable standard oligo for all replication events to act as a primer at the very beginning of the genome. While the other problem arises at the terminal end where the last base is added at a low efficiency in what is called the “last base addition problem” [26] due to the dinucleotide intermediates typically requiring two bases to extend. These two problems can be resolved if the RNA genome template was instead a circle, as there would be no beginning and have no end. Nevertheless, replication of a ssRNA circle to form a dsRNA circle would require efficient long tract RNA synthesis that, should it be successful, would cause the circle to become highly strained due to the high bending energy of dsRNA. Therefore, the early protocells likely would not have a circular RNA genome. 

Even with the stated problems above, Szostak and colleagues proposed a new model for replicating “primordial RNA genomes” through what they call a virtual circular genome [11]. This virtual circular genome contains multiple oligos that cover the entire circular RNA genome. Replication of the virtual circular genome will come after the annealing of the oligos and allow for templated addition of activated monomers, dimers, or trimers to allow for extension of the oligo. These oligos can then switch templates to allow for a continued elongation of the oligos to slowly replicate the entirety of the genome, allowing newly synthesized genetic material to pass on to progeny. Further, this mode of replication would also offer the ability for more than one copy of the RNA genome to be present in the same protocell, opening the door to the evolution of ribozymes. This evolution of RNA to form a ribozyme was studied in an in vitro assay [27] of a two domain RNA polymerase (B6.61) with the introduction of sequence variances at three distinct sites that then underwent selection. After 25 rounds of selection, what resulted were five families of polymerases able to bind a specific promotor and synthesize RNA. Together, this supports that the introduction of sequence variants into RNA can evolve to gain a function and that these RNA molecules can act as a ribozyme polymerase, and likely underlies the mechanism for RNA genome replication in the later protocells. 

These replicating protocells set the track for evolution that ultimately (after many evolutionary steps) arrived at where we are today. Though, we do not know the intermediate steps between the protocells to today, it is still possible that remnants of these protocells remain as viral pathogens. 

### 2.2. Single-Strand and Double-Strand RNA Virus Genomes

Generally, there are three classes of RNA viruses, and the key feature for separating the classes is based on the state the viral RNA genome (i.e., RNA packaged into the virion) is present [28,29]. The first class of RNA viruses maintains their genomes as double stranded (ds) RNA, examples include rotaviruses and reoviruses, while the other two classes are single stranded (ss). The single-stranded RNA viruses are further separated by the polarity of the genomic strand to be either negative (−) or positive (+), examples include for negative strand ssRNA viruses influenza, measles, mumps, rubella, Ebola, and Zika virus, and for positive stranded viruses include poliovirus, MERS-CoV. A further example of a positive-strand virus of particular interest is SARS-CoV-2. 

With these viruses being maintained differently requires different modes of replication (reviewed in [30]). Specifically, in positive-strand ssRNA viruses, RNA dependent RNA polymerase (RdRp) uses the positive genomic strand as a template to create a new negative strand copy (the anti-genome) that is subsequently used as a template to create large numbers of positive-strand viral RNA genomes (Figure 1). Alternatively, in negative-strand RNA viruses RdRp uses the genomic negative-strand as a template to create positive-strand antigenome, that also serve as mRNAs. RdRp subsequently uses the positive-strand anti-genome as a template to create large numbers of negative-strand viral RNA genomes (Figure 1). dsRNA viruses replicate their genome differently by generating positive-strand mRNAs (templated by the dsRNA genome) that are also used by RdRp as a template to create dsRNA genomes packaged into new virus particles (Figure 1). It should be noted that each virus type (even dsRNA viruses) relies upon multiple copies of ssRNA as intermediates of replication. This is important because, unlike dsRNA, the bases in ssRNA are not paired leaving the base exposed, therefore these ssRNA may be more prone to lesions such as base alkylation or chemical base deamination [31,32]. These lesions can stem from multiple sources (discussed at length in Section 5), but the major causes of lesions addressed in this review are the cytosine deaminases APOBECs and the adenine deaminases, ADARs. 

Both APOBECs and ADARs are part of the innate immunity and act as an antiviral by inducing base substitutions in viral genomes [33,34,35,36]. The polarity of the induced lesions in the resulting genomes depends on the virus class and their mode of replication. (Figure 1). In positive-strand ssRNA and in dsRNA viruses the predominant ssRNA species is the positive-strand, which is also an mRNA translated into viral proteins (Figure 1A,C). In negative-strand ssRNA viruses (Figure 1B), the negative RNA strand is more abundant. This bias in strand abundance can affect mutation accumulation bias, which may then be detected as mutation spectra strand bias.

## 3. Viral RNA Genome Rearrangements

Replication of RNA genomes in viruses is rarely perfect, often there is the introduction of errors [37] (discussed in Section 4) as well as the incomplete replication of the genome. When a genome is incompletely replicated it is either left unrepaired and degraded, or it is repaired by recombination with another RNA molecule, (reviewed in [38]). Viral recombination occurs at high rates between 2–20% recombination events per 100 nucleotides [39,40,41,42,43,44], and the rate of recombination is dependent on the fidelity of the RdRp [45,46,47,48]. Specifically, RdRps with a high fidelity are associated with a low recombination rate, while RdRps with low fidelity are associated with high levels of recombination [45,46,47,48]. This is due to the major RNA viral recombination mechanism being initiated by faulty, or incomplete, viral replication.

Replicative RNA recombination begins when the viral RdRp stalls during replication of the viral genome followed by the dissociation of the newly synthesized RNA molecule from the template that subsequently binds to another template where it is used as a primer to begin synthesis (Figure 2A), reviewed in [38]. The synthesis continues through the end of the template to complete the RNA molecule (Figure 2A) [41,42,49,50,51,52]. After RdRp stalling and dissociation, should the RNA find its homologous sequence in another identical RNA genome (Figure 2(Ai)), then the resulting genome will be identical to the previous template [41,42,49,50,51,52,53]. However, if the molecule that is utilized as the template is a completely different RNA molecule, which is possible due to the high number of mRNAs in the cell along with the possibility of infection by other RNA viruses, it will result in a chimeric RNA molecule containing parts of both RNA sequences (Figure 2(Aii)). This creates a novel viral genome, and should the recombination event include new beneficial genes, there may be the increased fitness of the virus. Such increase in fitness can be a major driver of evolution as well as may create new viral disease.

Non-replicative RNA recombination is a much rarer form of RNA recombination, occurring independent of RdRp, where two molecules are joined at their ends to create a chimeric molecule (Figure 2B) [54,55,56,57,58,59,60,61,62,63]. These events have been documented by modern sequencing approaches in viruses incapable of replication, but the mechanism for their formation is not yet known. Even so, a possible avenue for research may come from the many cell types that are able to recombine mRNA molecules through RNA splicing or RNA self-splicing to excise introns using small nuclear ribonucleoproteins (snRNPs) or through ribozymes [64,65,66]. Potentially, non-replicative recombination could use these mechanisms to join two different RNA molecules, instead of its more traditional function of excising introns. Regardless of the mechanism of recombination, what results is the formation of chimeric molecules that can become a novel viral RNA genome that may combine beneficial traits that helps with the virus’s overall fitness.

## 4. RNA Replication Errors

### 4.1. The RdRp’s Sequence Variation Effect on Replication Fidelity

RNA viral evolution has resulted in a diverse population of virus types that inevitably contain different combinations of genes within each virus, but present in all classes of viruses is RdRp. [67,68]. Though RdRp is conserved throughout RNA viruses, the overall RdRp sequence is highly variable, with some sequence conservation as low as 10% [69,70]. However, within this variable sequence of the RdRp, there are seven domains that contain key conserved residues. Specifically, the domains that are directly involved in ribonucleotide selection or catalysis contain conserved aspartate or lysine residues that when disrupted, greatly alters the RdRp activity [69,70]. The remaining sequence in the seven domains are not conserved to the same extent, but each specific sequence variation can modify RdRp function. Together, these sequence variations observed results in the variable RdRp replication fidelity, reviewed in [68], which is already orders of magnitude less accurate as compared to DNA replication. 

### 4.2. RdRp’s Replication Fidelity Impacts Viral Evolution

It was argued that the low fidelity of the RdRp drives the evolution of RNA viruses [71,72]. This idea was supported in multiples studies where viruses (influenza A virus, papilloma virus, foot-and-mouth disease viruses, Chikungunya virus, and human enterovirus 71) were exposed to nucleotide analogs (that increases the mutation rate, typically used as an antiviral strategy [73,74,75,76]) and after a few passages, a subpopulation emerged that became resistant through the acquisition of a mutant RdRp that has a higher replication fidelity [77,78,79,80,81]. Together, this suggests that a high mutation rate can mediate the formation of advantageous mutations that can drive evolution, but it also suggests that a higher replication fidelity can result in a more stable virus propagation. The latter notion is supported by viral strains that contained high fidelity RdRp variants continued stable propagation, however, they ultimately become attenuated [82,83]. 

With such a high mutation rate observed in viruses, there is a selection for smaller genomes because a larger genome would have more opportunities for the acquisition of a deleterious mutation resulting in “error catastrophe” [84,85,86]. Consequently, there is a balance of mutagenesis to be high enough to allow for adaptation, but low enough to be able to maintain a complex genome and prevent error catastrophe. This is believed to be a selection factor causing a tendency to limit of the size of the genome—for most RNA viruses to be around 15 kb in length [85,86]. Nevertheless, the Nidovirales family of viruses have RNA genomes upwards of 30 kb (maximum of 41 kb) [87,88] which is twice as large as a majority of viruses. A reason for the large viral genome size in Nidovirales remained unclear until the Gorbelyna group identified a sequence encoding a 3′ to 5′ exoribonuclease inside the SARS-CoV nsp14 subunit (called ExoN and referred to as so from here on), and speculated this is what could allow for the increase in genome size by proofreading RdRp errors and thereby reducing a chance of error catastrophe [89,90]. Subsequently, the 3′ to 5′ exoribonuclease function of ExoN was found in vitro and ExoN was also found to be essential for the viability of the alphacoronavirus HCoV-229E [91]. Similar experiments were conducted in ExoN-knockout mutants of two betacoronaviruses, MHV and SARS-CoV viruses [92,93], and found that the viruses were still viable, but to a much lower extent as compared to wild type viruses. Also, the ExoN-knockout mutants were deemed to have a “mutator phenotype” as they had a 15 to 21x increase, respectively, in mutations, as compared to wild type ExoN strains, approximately reaching the mutation frequency of other “non-nidovirales” viruses [92,93]. Together, this indicated that ExoN may act by proofreading RNA synthesis, which was later supported by the findings of that ExoN can excise mismatched nucleotides from a double-stranded RNA substrate [94,95]. 

## 5. Lesion-Induced Mutagenesis in Viral RNA Genomes

### 5.1. Environmental and Endogenous RNA Lesions and Modifications

Viral genomic RNAs as well as cellular RNAs are the subject to environmental and endogenous lesions. These lesions can result in RNA breakage or can block RNA, where the broken RNA genomes would be either lost or participate in recombination like events which can in turn produce rearranged genomes (see Section 3 and Figure 2). Like DNA, RNA base lesions and modifications can be caused by a variety of endogenous and exogenous agents (Table 1 and [31,32,96,97,98,99,100,101]). However, unlike for DNA, most base lesions in RNA cannot be repaired. The only known exception is for some alkylation products of cytosine and adenine bases which can be reversed to normal bases by a special class of oxidative demethylases—AlkB in bacteria or ALKBH family in humans [100,101,102]. Whether viruses utilize AlkB (or other similar proteins) to reverse alkylated bases remains unclear. The *Flexiviridae* plant viruses do encode for an AlkB sequence, gained through horizontal gene transfer, that when transcribed reverses alkylated bases within RNA and DNA, with a preference for RNA [103,104]. This suggests that AlkB does play a role in viruses, but a majority of viruses do not encode for an AlkB. It is possible, however, that viruses utilize the host’s AlkB instead of encoding for their own, but more work will need to be done to understand the role of AlkB to reverse alkylated bases in viruses. 

There have been 170 different RNA modifications identified in RNA across all species, with 60 of those being identified in eukaryotes [107]. Many of these RNA modifications have multiple functional consequences [99,108,109] and altogether are referred as the RNA-editome, or as the epitranscriptome. Investigations into the RNA modifications in viruses are still in its infancy, but even so, several enzymatic modifications of viral RNA bases have been reported (Table 1 [31,32,96,97,98,99,100,101]). Moreover, it is likely that there are additional modifications within RNA viruses are yet to be identified. Further, some of the identified RNA modifications have been reported to have physiological functions in RNA viruses, reviewed in [97]. Specifically, when uridines are replaced with pseudouridine in the Hepatitis C virus there is the disruption of the host interferon-β immune response, allowing for easier viral propagation [110]. Further, pseudouridine synthase is essential for the life cycle of *Flaviviridae* family of viruses indicating an essential role for pseudouridine in viruses [111]. The physiological functions of the N6-methyladenosine can affect viral replication where a low level of N6-methyladenosine leads to an increase in viral replication [97,105], while the increase of N6-methyladenosine results in less viral replication [112]. N5-methylcytosine, has been shown to be a positive regulator of viral replication [113] along with affecting the host immune response by binding to a recognition receptor but failing to initiate the antiviral cascade [114]. Lastly, hypoxanthine occurs in virus genomes after ADARs deaminate adenosine to inosine (hypoxanthine containing nucleoside) and impacts virus’ genomic sequence as inosine is copied as a guanine [115]. Though, these base modifications have been identified, their full mutagenic potential is yet to be determined. It is even not clear, if they are present in the full scale replicating viral genome or only in non-replicating viral mRNAs [105]. 

So far, only two kinds of enzymatic RNA edits—cytidine to uridine (C to U) by APOBEC cytidine deaminases (Figure 3) and adenosine to inosine (A to I) by ADAR adenosine deaminases (Figure 3) are known to be carried into copies of viral genomes resulting in C to U and A to G mutations respectively [116,117]. Henceforth, the next section will discuss APOBEC cytidine deaminases and ADAR adenosine deaminases as they may have the greatest impact on mutation accumulation in several human RNA viruses.

### 5.2. Base Substitution Mutagenesis in RNA Viruses

The introduction of base substitutions by APOBEC or ADAR is a common anti-viral strategy as excessive numbers of mutations is likely to result in inviable viral genomes [33,34,35,36]. However, base substitutions to a non-catastrophic level is an important source for viral evolution and population dynamics by introducing novel sequence variants [4,19,21]. These base substitutions are also a common avenue for viruses to escape the host’s adaptive immune system [33,34,35,36,118]. Thus, it is important to identify mechanisms underlying the generation of base substitutions in viral populations. Usual approaches to understanding the mutagenic mechanisms underlying genome mutations comes from a combination of knowledge accumulated in model studies as well as from agnostic documenting features of mutational spectrum that deviate from the spectrum expected if mutagenesis would be completely random [119,120,121]. Such “non-random” features of mutational spectra are also called mutational motifs (by analogy with musical motifs, which combine notes according to the rules of harmony) or mutational signatures (multiple set of features defining uniqueness of an object). This synthetic approach turned to be productive in revealing the mutagenic mechanisms in human cancers [119,120,121] through the use of cancer mutation catalogs—complete lists of de novo mutations in genomes of individual human tumors.

A similar strategy that was applied to human cancers can now be applied to several RNA viruses, especially because of the accumulation of extensive sequencing data for multiple viruses, most notably including SARS-CoV-2, which was a subject of a gigantic genome resequencing effort across the world [122] (see also URL https://www.gisaid.org/, accessed on 15 April 2021). However, since viral genomes are small, each genome would contain a small number, or even no mutations, so a mutation catalogue representative of a population (or a species) could only be built from sequences of multiple genomes. Building such a catalog must be done under the consideration of important factors to allow meaningful statistical evaluation of mutational spectra and signatures [116,117]. Firstly, an important point is that the mutations in the catalog must represent independent changes rather than a small number of events that are amplified through the development of a population, or through species evolution. A simplified example is shown in (Figure 4) where a viral population starting from a single genome accumulates mutations over nine rounds of copying (Figure 4) shows the importance of independent changes as many mutations in these genomes are identical, not because they occurred multiple times, but instead because they stem from a single genome that is then propagated.

Another important factor for the development of a mutational catalog is the knowledge of the original genome sequence from which all other genomes stem. This is required to identify the mutant alleles in every sequenced viral genome as well as to identify the directionality of the mutation events. The latter is especially important for defining mutational events across long term population dynamics or during species evolution. In the cases of identifying mutational signatures in mammalian RNA viruses, a single reference sequence for the entire dataset is not available, so the reference roles are assigned to the sequences in the nodes of phylogenetic trees built for genomes of isolates from a population [123] distant isolates of a single virus species [117,124,125] or from several related quasi-species [126].

Each node is taken as a surrogate of the reference sequence for the genomes in the same clade of a phylogenetic tree. With the surrogate reference sequence established, it can be utilized to develop a mutational catalog. This approach allowed for the detection of C to U changes as a prominent or even the major component of mutagenesis in a wide range of mammalian RNA viruses [127]. 

One more factor to account for in the analysis of mutagenesis results is strand bias of a particular change [116]. In viruses this bias will depend on a preference of a base modifying factor to single-stranded (ss) or to double-stranded (ds) RNA. It will also be affected by the kind of RNA forming genome of a virus: positive-strand ssRNA, negative-stranded ssRNA, or dsRNA (Figure 1A–C). The base changes shown on all panels of Figure 1 are the same as C to U changes expected from ssRNA-specific cytidine deaminases APOBEC and from A to I dsRNA-specific adenine deaminases ADAR (specific modification to the bases shown in Figure 4). Interestingly, the spectra and strand preference of the two most prevailing kinds of changes in hypermutated isolates of human vaccine-derived rubella virus corresponded to the prevailing C to U, and U to C, changes in genomic strand of this positive-strand ssRNA virus shown on Figure 1A [117]. These hypermutated viruses (up to 300 base substitutions in a 9 kb genome) were extracted from granulomas of different children with primary immunodeficiency. Each independent virus isolate stemmed from the attenuated rubella vaccine virus; whose known original sequence was used as a reference to build a mutation catalog from six isolates that contained 993 mutations. C to U, or A to G, changes were the major mutations in the catalog. Such a pattern of mutations in the rubella vaccine virus mutation catalog matches the signature of APOBEC cytidine deaminases and supports the idea that APOBEC enzymes are the major mutator. Recently, it has been established that APOBEC3A (A3A) enzyme has a preference to the unpaired parts (loops) of folded RNA structures in mRNAs of human tumors [128]. Importantly, C to U changes in the positive-strand in hypermutated rubella genomes were the only type of base substitution that showed a statistically significant density increase in predicted RNA-loops over stems—sequences with a potential for self-pairing [116]. This lends support for APOBEC mutation being the source for C to U changes, however both strand bias and unpaired loop preference could be the feature of any agent causing chemical deamination of cytidines in RNA. 

### 5.3. Mutation Signature Analysis in RNA Viruses

Deamination of cytosines not only occurs in RNA, cytidine deamination in DNA is one of the most frequent spontaneous changes and also has a preference to ssDNA [32], and all APOBECs with cytidine deaminase activity show clear preference to immediate nucleotide context surrounding deaminated cytidines in DNA. Specifically, APOBEC3G has a preference to cCn context (mutated nucleotide capitalized; n—any nucleotide), while APOBEC1 and all other members of APOBEC3 gene cluster prefer tCn deamination motif [129,130,131,132]. The preferred DNA deamination motif for APOBEC3A and APOBEC3B was even narrowed to tCa [133]. Unlike for DNA, detailed editing signatures in RNA are yet to be established. We therefore used APOBEC signature motifs established for DNA to evaluate the mutation spectra in a catalog compiled of hypermutated rubella genomes (Figure 5 shows example for the uCa→uUa motif). 

This method was initially developed for evaluating APOBEC mutagenesis in human cancers [134], however it allows statistical estimate of over-representation with any oligonucleotide motif in mutation datasets [133,135,136]. A fraction of mutations in an oligonucleotide motif among mutations of a given nucleotide is compared with the presence of the same oligonucleotide in the genomic context surrounding mutated bases (see also Figure 5 and legend). We found a high level of enrichment with APOBEC motif uCn and even greater enrichment with the narrower uCa motif which is also the most preferred DNA editing motif for APOBEC3A and APOBEC3B [117]. Unlike for APOBEC editing, there is only a multi-motif ADAR editing web-based Inosine-Predict score tool, which takes into account immediate nucleotide context for every guanosine position as well as the potential to form a secondary structure. This tool was developed for ADAR editing in mRNAs [137]. There was slight, albeit statistically significant increase in Inosine-Predict score for adenine positions involved in A to G mutations (U to C in complementary strand) mutations as compared to non-mutated positions of As (or Us) [117].

Altogether, APOBEC-like and ADAR-like changes represented 86% of the 993 mutations in the catalog from six hypermutated genomes of vaccine derived rubella virus. We then applied similar, but yet extended analytical and statistical evaluation approaches to evaluate mutation load and spectrum accumulated from over 30,000 SARS-CoV-2 genome sequences accumulated during first several months of pandemic [116]. In this analysis, we compared the spectrum and signatures with hypermutated isolates of rubella virus. The unique feature of this dataset is that the starting reference sequence is well defined [138], so each difference from the reference is a direct trace of a mutation event. However, some mutations were found in several thousands of isolates. We therefore used a set of non-duplicated mutations to represent the summary of mutational processes operating in pandemic SARS-CoV-2 population. We found that mutational processes with the same signatures that were revealed in hypermutated rubella isolates also may operate in the SARS-CoV-2 pandemic population. The main similarities between SARS-CoV-2 and rubella were: (i) the presence of APOBEC-like signature uCn to uUn in positive strand; (ii) frequent presence of ADAR like A to G and U to C (shown as in positive strand, will correspond to A to G in the negative-strand); (iii) preference of loops vs. stems for C to U mutations. Also specific to SARS-CoV-2 were the statistically significant enrichment with the two additional trinucleotide-centered signatures. Firstly, there was enrichment with mutations in cGn to cAn (reverse complement for nCg to nUg) which could reflect increased frequency of cytosine deamination in CpG motifs and C to U changes in the RNA negative-strand of dsRNA intermediate producing multiple copies of the positive-strand with the complementary G to A change (see example of negative strand mutagenesis in Figure 1A). Secondly, there was increased presence of G to U changes in the positive-strand, which could reflect C to A changes in the negative-strand. These changes could be caused by the increased formation of ROS-induced 8-oxoG within cells or during library preparation [139,140].

Recently, Adebali and colleagues performed an alternative approach to identify the mutational signature of SARS-CoV-2. They analyzed a large number of SARS-CoV-2 sequenced genomes organized in a phylogenetic tree [123]. Unlike in [116], this study used the nodes of the phylogenetic tree as a reference sequence to allow the identification of the mutations that dictate each node. Despite the two studies having used different strategies and approaches for creating datasets of independent mutation events in a large collection of SARS-CoV-2 genomes, the main categories of mutation preferences APOBEC-like, ADAR-like, CpG-like and apparent ROS induced mutations were similar. Overall similar conclusions about prevailing mutagenic sources and signatures were in works addressing intra-host variations of SARS-CoV-2 [141,142]. In summary, resequencing of RNA virus genomes suggested major mutational processes generating diversity that can lead to development of new virus forms. These studies also defined several questions and technical developments that should be addressed in near future.

## 6. Concluding Remarks and Future Questions

Emergence of new RNA viral quasispecies pathogenic for humans, especially the SARS-CoV-2 pandemic, triggered massive research efforts to all aspects of RNA virus mechanistic studies. Mechanisms underlying instability of RNA virus genomes are important for better prediction of their evolution, new pathogen emergence, and the development of antiviral drugs. Besides that, understanding biological and molecular mechanics that allows this group to flourish rather than be washed away with catastrophic error rates represents a fundamental question related to general mechanisms of evolution. Below are questions and technological applications that we anticipate being addressed soon.

### 6.1. Single-Molecule Sequencing Applied to RNA Virome

The sequence of a natural individual viral isolate is usually generated from a reference based or de novo alignment of multiple small Illumina reads, thus it does not reflect the variations of individual viral RNAs but instead is an average of the total population [143]. However, the recent combination of deep Illumina sequencing, and advanced bioinformatics, allows intra-host variations to be addressed in genomes of a single viral species during an acute infection period [141,142,144,145], so some interhost variation can be revealed. Even so, the combination of short reads with metagenomics does have its limitations as it relies on the building arbitrary contigs from short reads, so the entire genome cannot be assessed [146]. To overcome these short-read sequencing issues, there are two technologies carrying promise for studies of viral genome instability by generating long reads from individual polynucleotides, Oxford Nanopore Technology and Pacific Biosciences [147]. Each of the two platforms is plagued by a rather high sequencing error rate, but even with the current level of accuracy Oxford Nanopore was used for characterizing viromes [148,149,150]. Any further increase in sequencing accuracy may cause a revolution with viral genome instability research. While the field awaits this increase of accuracy of the core technologies, there is a way to reduce false positive mutation calls by adding unique molecule identifier (UMI) barcodes added by either limited number of PCR cycles or by ligation to increase the accuracy in both platforms [151].

### 6.2. Impact of RdRp Misincorporation and Proofreading onto Viral Mutation Rates

Low accuracy of RdRp, as compared with replicative DNA polymerases, led to the proposal that the major source of viral genome mutations is connected with replication errors [3,152]. Since many viruses have an exonuclease (ExoN or its homologs; see special section above) appearing to proofread RdRp misincorporation, it is important to collect more information about the impact of RdRp proofreading into prevention of hypermutation in RNA viruses. This would be approached by the modification of either, RdRp accuracy or ExoN capability by mutations and/or by endogenous or environmental factors. It is quite possible that the combination of such functional defects can lead to ultra-mutation phenotypes that would function similar to the synergistic hypermutation observed in cellular organisms when mismatch repair and DNA polymerase proofreading defects are combined [153,154,155]. Further, many antiviral drugs are chain terminating NTP analogs designed to preferentially affect chain extension by RdRp [73]. However, this chain termination can be counteracted by ExoN [90,156], so search for inhibitors of this enzyme is important for practical applications.

### 6.3. Are There RNA Repair Mechanisms besides AlkB Direct Reversal?

It is long known that RNA is more vulnerable to breakage as compared to DNA [157], and is at least just as susceptible as DNA to base lesions (Table 1 and references therein). However, unlike DNA, there is only one well established mechanism to repair RNA base lesions—direct reversal of alkylation. Currently there are no direct indications for the existence of other RNA repair mechanisms. Speculations can still be made based on structural similarity of RNA and DNA resulting in RNA being a substrate or a ligand for common DNA repair enzyme, e.g., RPA [158], but more research will be needed to reveal RNA repair mechanisms of they do indeed exist. 

### 6.4. Impact of Environmental RNA Lesions onto Viral Genome Instability

RNA, the same as DNA, is the subject for the lesions caused by environmental factors. DNA base lesions caused by environmental insults are well studied. They are often an impediment to replicative DNA polymerases and require specialized trans-lesion synthesis (TLS) DNA polymerases to successfully accomplish genome duplication. TLS polymerases are often error-prone and results in mutagenesis, while the lack of TLS can lead to genome rearrangements or to replication failure [159,160]. However, the same cannot be said about RNA genomes as, per current knowledge, there have been no model studies addressing the impact of environmental RNA damage on structural or sequence integrity of RNA genomes. This leaves a major gap in knowledge in how environmental RNA base damage is repaired, and whether it affects the stability of RNA genomes. This information is paramount for the understanding the dynamic world of RNA viruses.

### 6.5. Individual Host Impact of RNA Lesions onto Viral Genome Instability

Recent studies indicated that adenosine deaminases ADAR and cytidine deaminases APOBEC are the prevailing sources of base substitutions in several human RNA viruses including SARS-CoV-2, summarized in Section 5. Both of these enzyme types are the part of the innate immunity, which raises a question about RNA virus hypermutation within a single individual. Interestingly, in individuals with primary immunodeficiency in the adaptive immune system hypermutation of the vaccine-derived rubella virus was reported, which could be the reason of excessive activation of innate immunity and, consequently, resulting in the excessive activation of APOBECs [117]. Another important question is about the level of endogenous hypermutation of RNA viruses in species that may serve reservoirs for the occurrence of new quasispecies. Specifically, bats have been a known coronavirus reservoir that have multiple (4-7) APOBEC3 homologs while most of other mammalian orders have only one or two versions of APOBEC3 [161,162]. Therefore, studies into APOBECs within these organisms may reveal insights into the formation of novel viruses.

### 6.6. Experimental Models to Define Signatures of Environmental and Endogenous Mutagenesis in RNA

Defining diagnostic mutational signatures can develop into a multiprong scalable approach to understanding sources and mechanism of mutagenesis in RNA viruses. Mutational signatures turned to be a productive approach for another set of unstable genomes—human cancer. This could be a pure agnostic analysis of large datasets of genome instability catalogs [119,121,163], which can be also combined with prior mechanistic knowledge about different types of mutagenesis [120]. Another approach is to collect knowledge about mutational signatures in defined systems—mammalian [164,165] or microbial [133,135], and then utilize the information to build a specific statistical hypothesis for interrogating datasets of natural variants. High rates of mutation and relative ease of RNA virus genome sequencing can certainly make these approaches productive and scalable.

## Figures and Tables

**Figure 1 cells-10-01557-f001:**
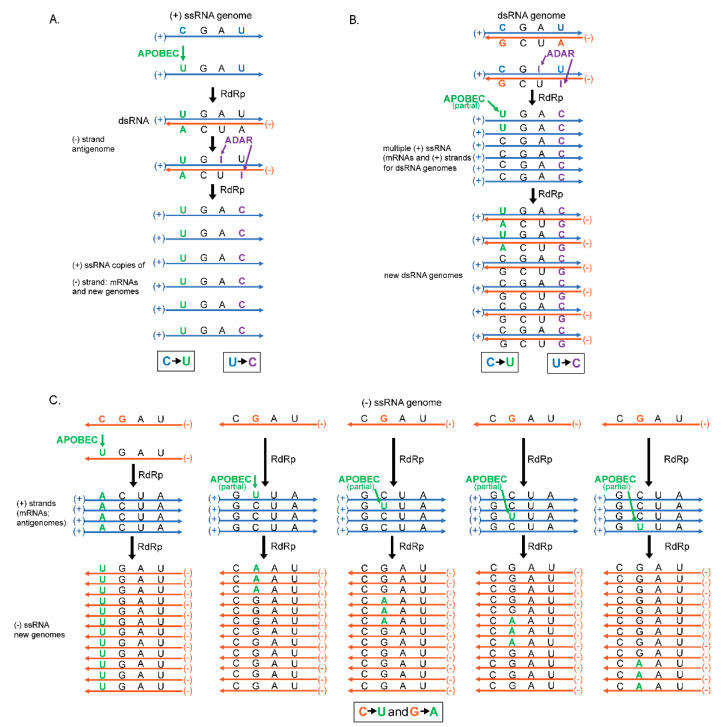
RNA virus genome type and mode of replication define mutation strand bias in the progeny. Presented are the modes of RNA viral genome replication and how mutagenesis with the ssRNA-specific cytidine deaminase APOBEC and the dsRNA-specific adenosine deaminase ADAR affect the genomes of the cell. Positive (+) strands are shown in blue. Negative (−) strands are shown in orange. Color codes, same as a strand color, are assigned to nucleotides that will be mutated in the next steps. APOBEC mutagenesis and resulting mutant nucleotides are shown in green. ADAR mutagenesis in dsRNA and resulting mutant nucleotides are shown in purple. Nucleotides that stayed not mutated in the progeny are shown in black. Predominant classes of mutations in progeny ssRNA genomes or in coding (+) strand RNA of dsRNA genomes is shown in boxes. (**A**) Viruses with positive (+) ssRNA genome. The infecting (+) strand RNA genome is used as a template by RNA dependent RNA polymerase (RdRp) to synthesize a dsRNA with both (+) and (−) strands. A single dsRNA molecule is subsequently used to generate multiple copies of (+) strand RNA transcripts and/or genomes. A single APOBEC-induced C to U change in the infecting genomic (+) strand ssRNA would amplify in all viral progeny (C to U mutations). An ADAR-induced A to I (inosine) change in the (+) strand dsRNA would not reproduce in genomes of viral progeny. In contrast, an ADAR-induced A to I change in the (-) strand dsRNA would be copied into multiple (+) strand RNA transcripts and thus be amplified in the viral progeny as U to C mutations in genomic (+) strand ssRNA. (**B**) Viruses with double-stranded (ds) RNA genomes. Multiple (+) ssRNA transcripts and/or genome precursors are generated by RdRp. Each (+) ssRNAs precursor is then used to generate a dsRNA genome. Only ADAR-induced A to I mutations in (−) strand are amplified into multiple dsRNA genomes via copies of (+) strands. Since there are multiple (+) strand intermediates, there is a chance of detectable level of C to U APOBEC-induced deamination in a fraction of (+) strands. (**C**) Viruses with negative (−) ssRNA genomes. Several (+) ssRNA transcripts and/or precursors of (−) ssRNA genomes are generated by RdRp that are then used to generate multiple (−) ssRNA genomes. (−) ssRNA genomes of infecting particles as well (+) ssRNA precursors can serve as a substrate for APOBEC mutagenesis. The change (C to U or G to A) recovered by sequencing progeny genomes would be defined by the strand which is deaminated by APOBEC. Multiple C to U mutant molecules will arise from a single deamination in the infecting (−) ssRNA genome. Smaller number of G to A changes would result from each deamination event in a (+) strand precursor, but since there may be multiple precursor copies (shown in the multiple columns), a number of these changes may be comparable with C to U changes.

**Figure 2 cells-10-01557-f002:**
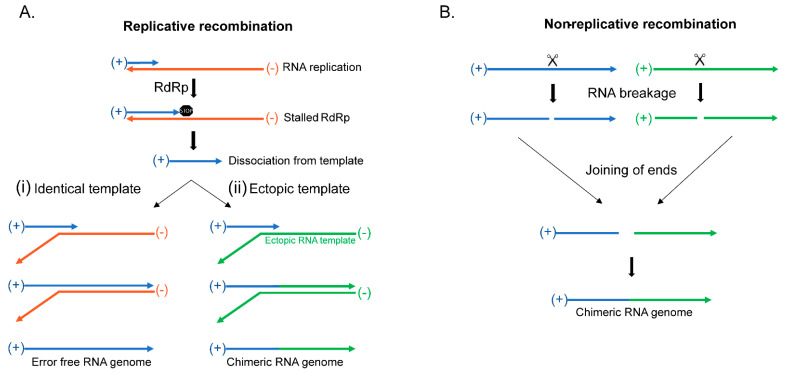
Viral RNA recombination. (**A**) Replicative recombination begins after incomplete RNA replication resulting in the dissociation from the template and rebinding with another RNA molecule to complete replication. (i) RNA template rebinding at a homologous location in an identical RNA template results in error-free recombination. (ii) RNA template rebinding in an ectopic RNA molecule creates a chimeric molecule. (**B**) Non-Replicative RNA recombination occurs through a yet unknown mechanism, which can involve breakage and joining of two different RNA molecules to create a chimeric RNA molecule.

**Figure 3 cells-10-01557-f003:**
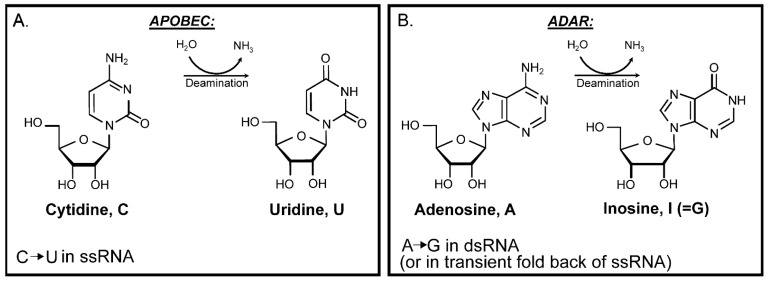
Enzymatic deamination of RNA nucleosides. (**A**) APOBEC cytidine deaminase. Deamination of cytidine in ssRNA generates uridine resulting in C→U mutation in the RNA virus genome. (**B**) ADAR adenosine deaminase. Deamination of adenosine in dsRNA or in folded and paired ssRNA (forming dsRNA) generates inosine, which after rounds of copying with RdRp is fixed as A→G mutation.

**Figure 4 cells-10-01557-f004:**
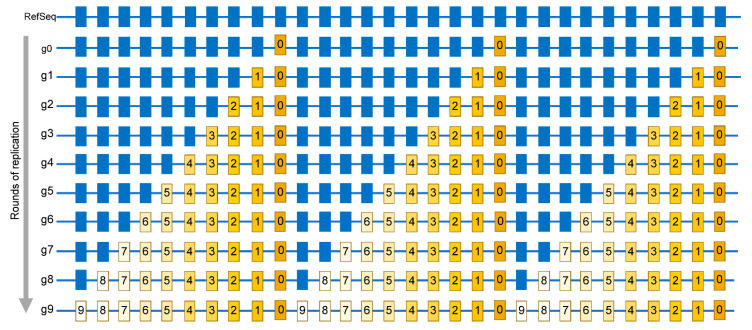
Simplified schematic of mutations accumulation in virus population. Mutations are identified by comparing a sequence of a viral isolate with a reference sequence (RefSeq). Individual positions where bases are mutated in at least one isolate are shown by rectangles. Blue rectangles are positions same as in RefSeq. g0—A genome of virus quasispecies starting a population that may already have some differences from RefSeq. g1—g9 rounds of replication generating additional mutations, which are numbered same as the generation in which a mutation event had occurred. Mutations occurring in later generations would be present in smaller fractions (reflected by the decreasing yellow color density) within the population. The entire set of independent mutation events would be described by the list in which every mutation is represented only once, regardless of the number of genomes where it is found. In this population, such a list is represented by the g9 genome.

**Figure 5 cells-10-01557-f005:**
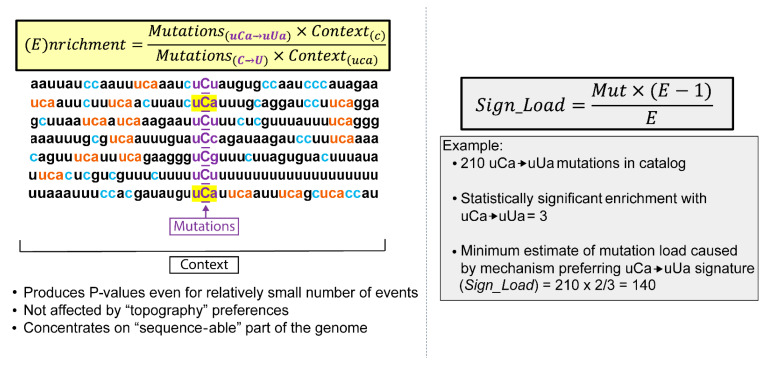
Trinucleotide motif-centered RNA mutational signature analysis. Shown is an example of an analysis for calculating the enrichment (E) and signature-associated mutation load (*Sign Load*) of uCa→uUa signature motifs in ssRNA, which are the two main outputs of trinucleotide motif-centered mutational signature analysis. Reverse complements are not included. Counted are all C→U mutations as well as all trinucleotide motif uCa→uUa mutations (5′ and 3′ flanking nucleotides shown in small letters; mutated C shown in capital letters). Also counted are all cytosines (c), represented in blue, and all motif-conforming trinucleotides (uca), represented in orange, in 41 nucleotide contexts centered around mutated cytosines. (E) values show the fold-difference between actual fraction of uCa→uUa mutations among C→U mutations in all trinucleotide motifs and the fraction of motif conforming trinucleotides (uca) among all cytosines (c) in the immediate vicinity of mutated cytosines. Counts used for enrichment calculation can be also used for calculating *p*-values in order to identify trinucleotide mutational motifs with statistically significant enrichment. Statistically significant enrichment values can be used for minimum estimates of a (*Sign Load*).

**Table 1 cells-10-01557-t001:** RNA base modifications.

Canonical Base	Modified Base	Possibility of Non-Enzymatic Generation	Enzymatic Generation	Enzymatic Reversal	Found in VIRUSES
Adenine	N1-methyladenosine (m1A)	SN2-alkylation	Methyltransferases	AlkB/ALKBH	Yes
Adenine	N3-methyladenosine (m3A)	SN2-alkylation	Not known	AlkB/ALKBH	TBD
Adenine	N6-methyladenosine (m6A)	Isomerization of m1A	Methyltransferases	AlkB/ALKBH	Yes
Adenine	Hypoxanthine base (Inosine ribonucleotide, I)	Low rate spontaneous deamination	Adenine Deaminases Acting on RNA (ADAR)	None	Yes
Uracil	Pseudouridine (Ψ)	Not known	Pseudouridine synthase	None	Yes
Guanine	N1-methylguanosine (m1G)	SN2-alkylation	Methyltransferases	AlkB/ALKBH	Yes
Guanine	N3-methylguanosine (m3G)	SN2-alkylation	Not known	Not known	TBD
Guanine	N7-methylguanosine (m7G)	SN2-alkylation	Methyltransferases	Not known	Yes
Guanine	O6-methylguanosine (O6mG)	SN1-alkylation	Not known	Not known	TBD
Cytosine	N1-methylcytosine (m1C)	SN2-alkylation	Not known	Not known	TBD
Cytosine	N3-methylcytosine (m3C)	SN2-alkylation	Not known	AlkB/ALKBH	TBD
Cytosine	N5-methylcytosine (m5C)	Not known	Methyltransferases	Not known	Yes
Cytosine	Uracil	Spontaneous deamination	APOBEC/AID	None	Yes

Table 1 contains a summary of information compiled from [31,32,96,97,98,99,100,101,105,106].

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
