# Peer review of "From RNA World to SARS-CoV-2: The Edited Story of RNA Viral Evolution"

_cells, 2021, doi:10.3390/cells10061557_

Round 1
Reviewer 1 Report
Kockler ZW and Gordenin DA have put together an outstanding review on the evolution of RNA viruses, especially under the lights of RNA modifications and editing. They discussed how editing deaminases are incorporated in generating different mutational spectra in in different viruses, touching the topics of recombination and further the impact of RNA modifications. All the topics addressed in the review are nicely and cohesively connected, however there are a few things that I think the authors should clarify or change within the text. Here, I provide my specific points:
- Section 2.2 starting in line 142: great introduction about ss and dsRNA viruses, at the end of the section or within the text of it, please give examples of ssRNA and dsRNA viruses. Here’s the opportunity to introduce SARS-CoV-2 a little better.
- Lines 161-163: please, give a reference for the statement that ssRNA viruses are not protected by hydrogen bonds, while dsRNA viruses are.
- In line 164 you cite Figure 1; the main text is mainly about polarity etc which is depicted in the figure, but what is also depicted in the figure and not discussed in the main text (while it is discussed further), is the ADAR and APOBEC impact. Please, introduce the ADAR and APOBEC impact here and discuss or pinpoint where you discuss it further down within the main text.
- Line 356: please replace the “references in footnotes” with the actual references.
- Within the section 5.1, please discuss a little bit more about the RNA modifications, other than ADAR and APOBECs, especially because you summarize them within the Table 1 and I believe not enough focus has been given on them throughout the manuscript.
- Table 1: It is a great idea to summarize the different modifications and their writers. I however think that this table is not up to date. For instance, for m1A we now know which one is the enzymatic writer of this modification. Please check for instance this review: Delaunay, S., Frye, M. RNA modifications regulating cell fate in cancer. Nat Cell Biol 21, 552–559 (2019). https://doi.org/10.1038/s41556-019-0319-0. I would also advise the authors to update the literature on RNA modifications.
- Line 384: They are also a common … immune system: I understand the point – it’s basically common sense – but I would urge the authors cite the appropriate reference. It is a bold statement.
- Within the section 5.2, I reckon it’s crucial to address around the lines 474-494 (or after this paragraph) the fact that ADARs and APOBECs were originally thought to be antivirals (which they are), but how could this be bypassed under the prism of evolution? I would encourage you to discuss this.
- Line 474: from guanine (G) to inosine (I): do you perhaps mean Adenosine (A) to Inosine (I)?
- Figure 3: This representation is awesome. Could you perhaps make one of the three mutational cascades “imperfect”? It get the point, but it gives the wrong impression that within a given population of viruses, the latest present (here g9) will be 100% different than the original (RefSeq), which I don’t believe that this is what you mean.
- Line 506-507: It is important to clarify. I believe the authors mean that all APOBECs that present deamination activity (such as APOBEC1, APOBEC3s) can deaminate DNA in motifs etc. The "All APOBECs" would include APOBEC2 or APOBEC4, which are not deaminases.
Author Response
attached is the word file with cover letter and point-by-point responses to both reviewers

Reviewer 2 Report
This review by Kockler and Gordenin is focused on RNA integrity including insults that occur during replication of RNA viruses and from endogenous and exogenous sources of damage to viral and cellular RNA. RNA integrity and evolution is of increasing interest in the scientific community, and viruses represent an essential model for the study of RNA modifications, damage, and repair. Given the recent pandemic caused by SARS-CoV-2, an RNA virus, this review provides topical information.
Overall, the review is fairly dense and complex. The authors would do well to limit the scope of topics discussed and to focus on pertinent points. There is a significant amount of space given to seemingly less relevant subjects such as APOBEC enzyme activity on DNA genomes. The manuscript is also asymmetric in the amount of primary data discussed in some sections compared to others, this should be balanced to improve readability. Similarly, several of the figures are either overly complex or not entirely relevant. This is an important subject and will be well received in the current scientific environment, but the following major points should be addressed:
- The entire manuscript requires editing for brevity and clarity. Several sections are redundant with one another (for example section 4.1 reintroduces RdRp which was already introduced at length in section 2.2). Additionally, several sections are overly written and would benefit from more concise writing – specifically, sections 1, 4.1, 4.2, 5.2, and 6.1.
- The discussion of RNA world is complicated and should be more concisely summarized. The long discussion of RNA replication is of unclear relevance to the remainder of the review.
- Figure 1 needs to be simplified. The purpose for including APOBEC/ADAR at that point in the manuscript is unclear.
- The discussion on ExoN includes far more preliminary data than the rest of the review – this section should be made more consistent with the remainder of the manuscript. The ExoN discussion is also of unclear relevance to RdRp or viral evolution so the title of the section should be clarified or this discussion moved to a distinct section. The last paragraph of section 4.2 should probably be excluded since it is a detailed description of primary data that is not relevant to the overall point of the review.
- Table 1: which RNA base modifications are relevant in virus genomes? This should be clearly stated. Does ALKBH act on virus genomes?
- The points of figures 3 and 4 are unclear, they likely can be excluded.
- Section 5.2 is lacking citations, especially page 15 lines 452-485.
- The extensive discussion in section 5.2 on APOBEC activiy on DNA is not relevant and should be cut to a brief summary.
- Figure 5 is very confusing for a non-computational biologist. Is it meant to be a pro/con of motif analysis? Or a primer on how to do motif analysis?
- Page 19 lines 546-589 reads as a results/discussion section of reference 132. Cut to a clear point or two that is relevant to the review subject. One option is to focus on different ref seq methods used by references 125 and 132 that ultimately resulted in the same findings.
Minor points:
- On page 4, the authors seem to suggest that dsRNA genomes are less prone to genome instability than ssRNA genomes, but the section needs a concluding statement.
- Section 3 uses the full name of RNA dependent RNA polymerase, whereas in prior sections it was abbreviated to RdRp.
- The end of the first paragraph of section 4.1 is a nonsequiter.
- Viruses listed in section 4.2 need full names, not just acronyms.
- Page 12, line 375-378 needs a citation.
- The second paragraph of section 5.2 is off-topic and should require only a brief, if any, introduction. Lines 422-430 of section 5.2 requires significant clarification.
- The title of section 6.1 should be more relevant, and the section should be shortened to only the very relevant conclusions/future directions of RNA sequencing.
- Section 6.4 requires clarity – is this a comment on environmental RNA lesions or translesion synthesis repair of DNA? Same for section 6.5 – is this comment just about APOBEC? If so, this should be reflected in the title. Redundancy with prior sections on APOBEC should be avoided.
In general, this review requires major revisions to ensure a cohesive, clear, and readable manuscript, but covers and important subject which of interest to the current pandemic and will positively contribute to the field.
Author Response

(The authors gave the same response as above.)
